# Outbreak of Feline Sporotrichosis with Zoonotic Potential in the Seventh Health District of Maceió-AL

**DOI:** 10.3390/jof10070473

**Published:** 2024-07-10

**Authors:** Ana Paula de Castro Pires, Júllia de Oliveira Siqueira, Maria Rafaela Pereira Gomes, Janaína André da Silva, Alisson Luiz da Costa, João Paulo de Castro Marcondes, Aryanna Kelly Pinheiro Souza

**Affiliations:** 1Sector of Health and Biological Sciences, Centro Universitário Cesmac, Maceió 57051-160, Alagoas, Brazil; 2School of Pharmacy, Centro Universitário Cesmac, Maceió 57051-160, Alagoas, Brazil; 3School of Veterinary Medicine, Federal University of Alagoas, Maceió 57700-970, Alagoas, Brazil

**Keywords:** *Sporothrix* spp., cats, dermatitis

## Abstract

Sporotrichosis is a mycosis with zoonotic potential caused by species of *Sporothrix*. Once thought rare in northeastern Brazil, the disease has recently been spreading, leading to an emergency health issue. In this paper, we describe an outbreak of feline sporotrichosis in the Seventh Health District of Maceió-AL. We collected samples from 23 domiciled and non-domiciled felines without regard for age, breed, sex, and neutering state. Skin samples were analyzed cytologically under a light microscope and seeded onto Sabouraud dextrose agar at 25 °C for from 15 to 30 days. Fifteen of the twenty-three cats with suspected skin lesions were positive for *Sporothrix* spp. on either cytological or microbiological evaluation. Most of the infected cats were male, young adults, non-neutered, with free access to external areas, and living in environments with poor sanitation, a high population density, and an accumulation of garbage and organic matter. Three owners were bitten or scratched by infected cats and subsequently developed suspicious cutaneous lesions suggestive of sporotrichosis. The epidemiological features of feline sporotrichosis in the outbreaks of Maceió seemed to share similarities with the data obtained from outbreaks in current hyperendemic areas. Identifying geographical sites of infection and providing compulsory notification of the disease is essential for avoiding an epidemic in Alagoas.

## 1. Introduction

Sporotrichosis, a zoonotic subcutaneous implantation mycosis, has been increasingly reported globally. This disease is caused by species belonging to the Sporothrix complex. These causative agents are thermodimorphic and saprophytic fungi, commonly found in soil and decaying plant matter in subtropical and tropical regions [1,2,3].

The disease has historically been described as an occupational mycosis. Farmers, gardeners, and other professionals constantly exposed to the natural environment of the fungi were at a higher risk of infection, and zoonotic sporotrichosis was rarely reported worldwide. However, the process of rural exodus and urban verticalization changed the disease’s profile, and nowadays, cat-transmitted sporotrichosis is the most prevalent infection route in endemic countries [3,4].

In 1998, Rio de Janeiro (Brazil) experienced an alarming increase in cases of human sporotrichosis transmitted by infected cats. Since then, Brazil has been facing a geographic expansion of zoonotic sporotrichosis [4]. Rio de Janeiro evolved into a hyperendemic area, and several municipalities in south and southeast Brazil are dealing with severe public health issues [5].

Health professionals previously believed that human and feline sporotrichosis was rare in northeastern Brazil, and few studies addressed the topic. In the past five years, however, case and outbreaks of zoonotic sporotrichosis have been reported in the states of Pernambuco, Paraíba, Salvador, and Rio Grande do Norte [6,7,8,9]. In the State of Alagoas, apart from empirical data, only one case of feline sporotrichosis associated with human infection has been reported since 2014 [10].

Only a few Brazilian municipalities perform a compulsory notification program for zoonotic sporotrichosis. Therefore, the disease’s geographic expansion, epidemiologic aspects, morbidity, and mortality rates remain unknown in the national sphere [5]. A comprehensive sporotrichosis control program is beyond the sphere of public health, and a One Health approach is necessary to control the disease in humans and mammals [11]. We aimed to describe an outbreak of feline sporotrichosis with zoonotic transmission in the Seventh Health District of Maceió-Alagoas, Brazil.

## 2. Materials and Methods

### 2.1. Ethics Statement

The Animal Research Ethics Committee of the Centro Universitário Cesmac, AL, Brazil, approved this research under the CEUA 11A/2019 protocol. The cat owners signed permission authorizing the procedures and agreed to share clinical and epidemiological details of the cases.

### 2.2. Study Site

This study was conducted in Maceió—the capital of Alagoas, northeastern Brazil. Its typical climate is hot and humid with a low thermal amplitude and high pluviosity in the fall and winter, notably between April and August [12]. Maceió houses approximately 29.94% of the population of Alagoas and extends for 509,552 km/m^2^. Following the geographical organization designed by the Brazilian Unified Health System, the 51 neighborhoods of Maceió are subdivided into 8 health districts (Figure 1). The Seventh Health District covers five neighborhoods—“Cidade Universitária”, “Clima Bom”, “Santos Dumont”, “Tabuleiro dos Martins”, and “Santa Lúcia” [13].

### 2.3. Study Design, Cats, and Sample Collection

Between 2020 and 2022, we obtained samples of 23 domestic felines (*Felis catus*) presenting single or multiple ulcerative and exudative skin lesions suspected of sporotrichosis without regard for gender, age, neutering state, and breed. Data on the address, age, gender, breed, neutering state, access to external areas, human and animal cross-contamination, and morphological aspects of the lesions were obtained and recorded on a standard clinical file.

### 2.4. Cytologic Diagnosis

Cytologic specimens were obtained through impression (imprint), scrapping, or fine-needle aspiration cytology techniques, as instructed by Raskin and Meyer [14]. The slides were air-dried, fixed in methanol, and stained with a commercial fast-staining Panotico kit (Instant Prov; Newprov^®^, Irvine, CA, USA). The slides were analyzed by light microscopy under 400× magnification.

### 2.5. Fungal Culture

Samples of skin lesions were collected with a sterile swab and conditioned in a transport medium (Stuart, Santa Fé, Argentina). The samples were seeded onto Sabouraud dextrose agar and incubated at 25 °C for 7–15 days. *Sporothirx* spp. was identified by morphology and conversion to the yeast phase at 35 °C.

## 3. Results

The first outbreak of feline sporotrichosis occurred in November 2020 in the house complex “Village II” (“Cidade Universitária” neighborhood). A local woman noticed the onset of skin lesions on her cat and requested veterinary assistance to investigate the condition. During the interview to collect the history and clinical data of the patient, the woman disclosed that a second cat in her household and a few stray and semi-domiciled cats in her block were presenting similar skins lesions.

Our team then performed two visits in the area and collected samples from 11 cats presenting multiple exudative and ulcerative skin lesions. Seven cats were positive for *Sporothrix* spp. either on cytological or microbiological examination. Moreover, three owners reported being bitten or scratched by an infected cat. These people were given orientation on the risks of zoonotic sporotrichosis and were referred to a hospital specialized in tropical diseases to receive proper diagnosis and treatment. Data on the positive cats and geographical location of the outbreak can be seen in Table 1 and Figure 2, respectively.

A few months later, in January 2021, the zoonotic surveillance unit (UVZ) of Maceió contacted our team to assist in a possible occurrence of feline sporotrichosis in a housing complex known as “Maceió I”, also located in the “Cidade Universitária” neighborhood. We collected samples of 12 cats with suspected skin lesions and later confirmed the diagnosis of sporotrichosis in 8 of them (Table 2).

Considering the 15 positive felines in the two outbreaks, all the cats were of a mixed breed, 12 were males and 3 were female. The informed age ranged from three months to eight years old, with a mean of 4.8 years. In 10 cats, the exact age could not be established, and those were considered adults (1–6 years) based on dental devolvement and the owner’s history of adoption. Only two cats had been neutered at the time of diagnosis. According to information provided by the owners, all cats had free access to external areas near the household. The clinical history of territorial fights with neighboring felines was recurrent. Five cats were non-domiciled and were rescued with suggestive cutaneous lesions.

Clinically, the infected felines presented focal and multifocal-to-coalescent nodular or ulcerative, exudative, hyperemic, and hemorrhagic skins lesions distributed on the face (Figure 3A), external and internal nose, limbs, and interdigital area. Less common sites included the tail, cervical, and dorsal areas. Mucosal involvement, cartilage destruction, “clown nose”, and respiratory impairment were frequent features associated with nasal lesions. 

Cytopathologic examination of the positive cats revealed moderate-to-intense pyogranulomatous inflammation with degenerated neutrophils and pink protein deposits on a bloody background. Macrophages were distended and presented numerous intracytoplasmic oval-to-elongated, cigar-shaped, and thin-walled yeast cells with a clear cytoplasm and peripheric nuclei, measuring from 3 to 5 µm (Figure 3B). Similar structures were also seen in the background. Secondary bacterial contamination was common. The samples seeded onto Sabouraud dextrose agar yielded brown-to-black colonies featuring septate hyphae that were from 1 to 2 μm wide, with conidiogenous cells arising from undifferentiated hyphae forming conidia in groups on small, clustered denticles—daisy-petal pattern (Figure 3C,D).

## 4. Discussion

In the last 20 years, Brazil has experienced a geographical expansion of zoonotic sporotrichosis [11]. Sporotrichosis in northeastern Brazil was rare compared with hyperendemic states such as Rio de Janeiro; isolated cases of the disease were reported occasionally, mostly associated with rural occupations. This epidemiologic profile started to change in 2014 when the first significant outbreak of feline sporotrichosis was reported in Pernambuco [8].

During the next few years, zoonotic sporotrichosis turned into a notifiable disease in Pernambuco, Bahia, and Rio Grande do Norte. However, the data on zoonotic sporotrichosis in northeastern Brazil was likely underestimated. In Alagoas, for instance, only one case of zoonotic sporotrichosis was reported [10].

In our study, we investigated 23 domicile and non-domicile cats living in two house complexes in the Seventh Health District of Maceió; 15 cats were positive for *Sporothix* spp. on either cytology or microbiological cultures. Most of the infected cats were male, young adults, non-spayed, and roamed freely outdoors, similar to the epidemiological profile reported by Silva et al. [8] in Refice-Pernambuco. Young males and fertile cats are predisposed to infection with *Sporothrix* due to their natural behavior of fighting other animals for food, females, and territory. Those habits facilitate skin inoculation of the fungi and favor its spread in the environment [15].

Out of the 15 cats diagnosed with sporotrichosis in our study, 13 tested positive in both cytopathology and fungal culture, 2 were positive in fungal culture but negative in cytopathology, and 1 was positive in cytopathology but negative in fungal culture. Isolation and culture of *Sporothrix* spp. from clinical samples is recognized as the gold standard for diagnosing feline sporotrichosis. As detailed by Silva et al. [8], the filamentous form of *Sporothrix* spp. is characterized by delicate, septate hyphae with conidia arranged in rosettes or in a “daisy” pattern on conidiophores when observed at 25–30 °C.

However, despite its high sensitivity and specificity, fungal culture can occasionally yield false-negative results due to factors such as inadequate sampling and contamination with opportunistic agents [3]. In this context, cytology emerges as a valuable complementary diagnostic tool. In this study, we found that the characteristic cigar-shaped yeast cells of *Sporothrix* spp. were readily observable in the cytological preparations from most of the cats that tested positive. França et al. [15] asserts that cytology is a simple, fast, and cost-effective diagnostic tool that is particularly suitable for use in cats. This is largely because the exudative lesions of sporotrichosis in cats are typically rich in microorganisms, facilitating a rapid and straightforward presumptive diagnosis following slide staining.

During the anamnesis and clinical evaluation of the feline patients, three owners disclosed being bitten or scratched by a wounded cat, developing ulcerated skin lesions shortly after the incident. Our team lectured the owners on the zoonotic potential of sporotrichosis and encouraged those people to seek a specialist in our local Hospital for Tropical Diseases. Although our team was not able to follow-up the suspicious human sporotrichosis cases, nor did they have access to the laboratory tests confirming or refuting this hypothesis, it is undeniable that people living in the areas of the outbreaks are at risk of contracting zoonotic sporotrichosis. Gremião et al. [11] claim that in areas where only feline sporotrichosis is notifiable, the zoonotic condition goes unnoticed, and an increase in feline cases usually follows an increase in human cases. In a study conducted in Rio Grande do Norte, out of the 122 patients admitted to a tropical reference disease hospital, 115 had previous contact with cats, and 81 reported being bitten or scratched by an infected cat [9].

The geographical expansion of sporotrichosis has been associated with social and economic inequality, unemployment, poverty, urban agglomeration, lack of sanitation, and poor public health services [9]. In our study, the two outbreaks occurred in the Seventh Health District of Maceió. The Seventh Health District of Maceió includes the neighborhoods “Cidade Universitária, “Santos Dumont”, “Clima Bom”, Tabuleiro dos Martins”, and ‘Santa Lucia”. It is an area that extends for 44,100 square kilometers and it is inhabited by 268,739 people (the most populated district in Maceió), with an estimated geography density of 6093.85 habitants per square kilometer [13]. According to the data provided by the Brazilian Statistic and Geographical Institute, most people in the Seventh Health District earn less than the minimum wage and live in houses with a backyard [16]. Sanitation is poor, many streets are not asphalted, and cats, dogs, chickens, and pigs roam freely, surrounded by garbage and organic matter [17].

A study conducted in São Paulo demonstrated that the higher the social vulnerability of a population, the higher the incidence of sporotrichosis in this area [16]. Alzuguguir et al. [18] obtained similar results in a hyperendemic area of Rio de Janeiro. They noted a relationship between a high incidence of sporotrichosis, a low per capita income, and urban agglomeration. On the other hand, sanitation and garbage collection did not play a role in the case distribution.

Regardless, deficits in environmental sanitation contribute to several neglected tropical diseases. Lack of sanitation in affected areas may facilitate fungus transmission due to potential soil contamination and an increase the transmissibility of sporotrichosis to humans and animals [19]. In this sense, more comprehensive studies are necessary to establish if socioeconomic aspects such as the ones present in the Seventh Health District play a role in the geographical distribution of sporotrichosis in Maceió.

Outbreaks of zoonoses in urban areas require targeted strategies for their prevention and control. In the public and governmental sphere, the recommendation is the population’s education on transmitting sporotrichosis, programs to limit feline reproduction, treating cats, implementing basic sanitation, regular garbage collection, and cleaning of vacated lots. Responsible ownership of cats is essential in the private sphere and includes limiting outdoor access, cleaning shelters, and reducing the abandonment of infected cats [11]. It is important to note that to date, sporotrichosis is not a notifiable disease in Maceió.

## 5. Conclusions

The epidemiological characteristics of feline sporotrichosis during the outbreaks in Maceió appear to resemble those observed in current hyperendemic areas. Identifying the geographical locations of infection and implementing mandatory disease reporting are crucial steps to preventing an epidemic in Alagoas.

## Figures and Tables

**Figure 1 jof-10-00473-f001:**
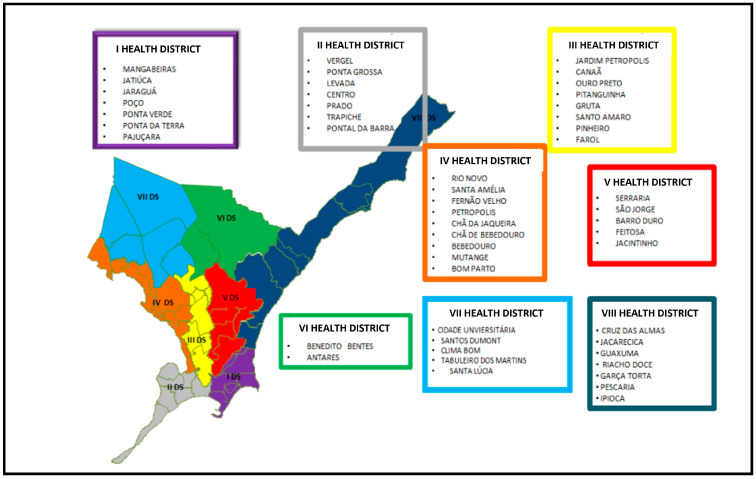
Neighborhood distribution by health districts in Maceió-Alagoas.

**Figure 2 jof-10-00473-f002:**
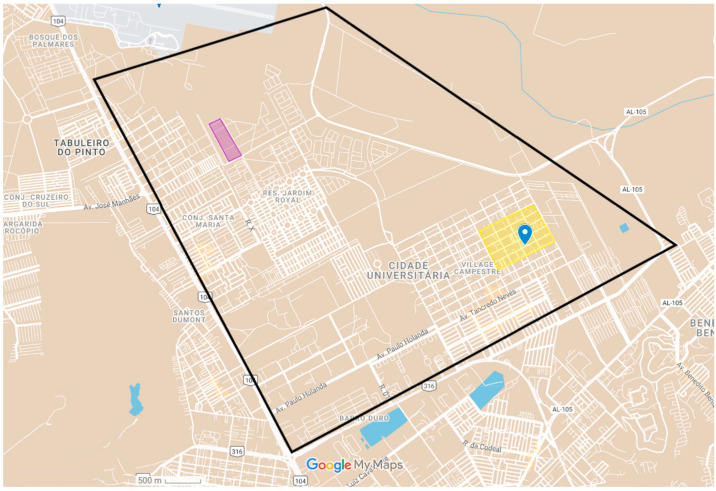
Distribution of the outbreaks of sporotrichosis in Maceió-AL. Within the black lines is the Cidade Universitária neighborhood. The yellow square shows the location of the first outbreak, and the purple rectangle delimits the area of the second outbreak.

**Figure 3 jof-10-00473-f003:**
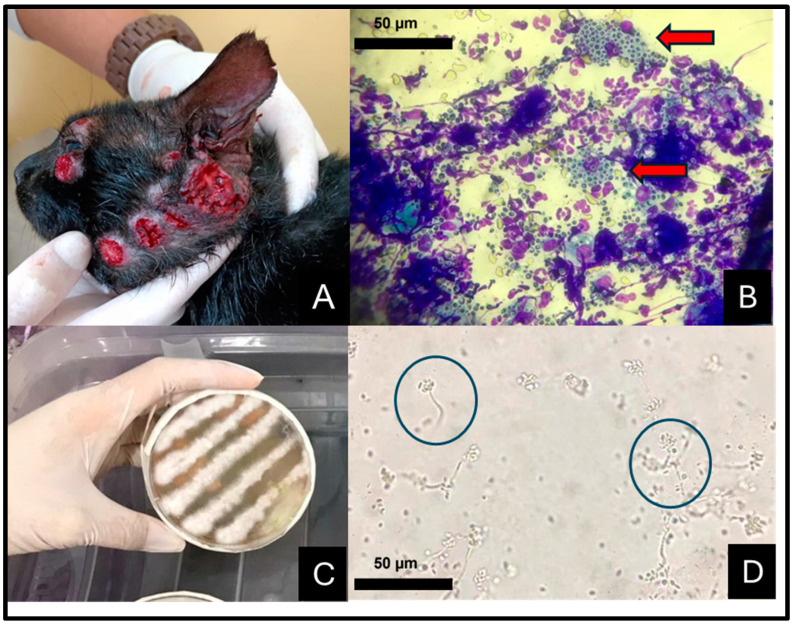
Feline sporotrichosis in the Seventh Health District of Maceió-AL, northeastern Brazil. (**A**) Cat presenting multiple ulcerative, exudative, and hemorrhagic lesions on the face. (**B**) Cytologic sample exhibiting intense pyogranulomatous inflammation and intra-histocyte yeast structures (red arrows). Fast panoptic stain, 40×. (**C**) Yeast formed seven days following inoculation on Sabouraud dextrose agar at 37 °C. Obj. 40×. (**D**) Tape prep of fungal growth in Sabouraud agar stained with lactophenol cotton blue showing colonies of undifferentiated hyphae in a daisy-petal pattern (blue circles). Obj. 40×.

**Table 1 jof-10-00473-t001:** Epidemiological data and test results of cats positive for *Sporothrix* in the housing complex Village II.

ID	Breed	Gender	Age	Neutering Status	Cytology	Culture
V1	Mix-breed	Male	Adult	Non-neutered	(+)	(+)
V2	Mix-breed	Male	Adult	Non-neutered	(+)	(+)
V3	Mix-breed	Male	Adult	Non-neutered	(+)	(−)
V4	Mix-breed	Female	4 years	Non-neutered	(+)	(+)
V5	Mix-breed	Male	4 years	Neutered	(+)	(+)
V6	Mix-breed	Male	5 years	Neutered	(+)	(+)
V7	Mix-breed	Female	Adult	Non-neutered	(+)	(+)

(+) Positive; (−) negative.

**Table 2 jof-10-00473-t002:** Epidemiological data and test results of cats positive for *Sporothrix* in the housing complex Residencial Maceió.

ID	Breed	Gender	Age	Neutering State	Cytology	Culture
M1	Mix-breed	Male	Adult	Non-neutered	(+)	(+)
M2	Mix-breed	Male	Adult	Non-neutered	(−)	(+)
M3	Mix-breed	Male	Adult	Non-neutered	(−)	(+)
M4	Mix-breed	Male	Adult	Non-neutered	(+)	(+)
M5	Mix-breed	Male	Adult	Non-neutered	(+)	(+)
M6	Mix-breed	Male	8 years	Non-neutered	(+)	(+)
M7	Mix-breed	Female	Adult	Non-neutered	(+)	(+)
M8	Mix-breed	Male	3 years	Non-neutered	(+)	(+)

(+) Positive; (−) negative.

## Data Availability

The original contributions presented in the study are included in the article; further inquiries can be directed to the corresponding author.

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
