# Peer review of "Outbreak of Feline Sporotrichosis with Zoonotic Potential in the Seventh Health District of Maceió-AL"

_jof, 2024, doi:10.3390/jof10070473_

Round 1
Reviewer 1 Report
Comments and Suggestions for Authors
The authors describe an outbreak of feline sporotrichosis and zoonotic transmission in northeastern Brazil. Feline sporotrichosis has been well-documented Brazil and while this work is not particularly novel, it expands the geographic area of concern in Brazil. A few questions or comments for the author's consideration:
1. I don't think Figures 2-4 are all necessary. Consider retaining Figure 2 and delete Figures 3 and 4.
2. Figure 5 - for quadrant B, please add an arrow or two pointing to an example of an intra-histocytic yeast. Quadrant C legend should read 37oC. Quadrant D, the description in the figure legend is a bit confusing. Is this a wet-prep or a tape-prep with what stain prepared from a colony that grew from a sample seeded onto Sabouroud's agar?
3. line 184 - should be bitten instead of bitter.
4. line 195 - "have an income down the minimum wage" needs revision since "down" doesn't make sense here. Would "at" be better?
5. line 206 - Sporotrichosis instead of Sporothicosis.
6. Table 2 - Sample ID M7 should read No instead of N0 in the right-hand column.
Author Response
- I don't think Figures 2-4 are all necessary. Consider retaining Figure 2 and delete Figures 3 and 4 - I have kept figure 2 as suggested.
- 2. Figure 5 - for quadrant B, please add an arrow or two pointing to an example of an intra-histocytic yeast. Quadrant C legend should read 37oC. Quadrant D, the description in the figure legend is a bit confusing. Is this a wet-prep or a tape-prep with what stain prepared from a colony that grew from a sample seeded onto Sabouroud's agar? It was a tape pret stained with Lactophenol cotton blue. I have added it in the legend.
- 3. line 184 - should be bitten instead of bitter - okay
- 4. line 195 - "have an income down the minimum wage" needs revision since "down" doesn't make sense here. Would "at" be better? -I meant this people per capita income is inferior to the minimum wage established in Brazil. I have rephrased it; please let me know if it is more clear now
- 5. line 206 - Sporotrichosis instead of Sporothicosis. - okay
- 6. Table 2 - Sample ID M7 should read No instead of N0 in the right-hand column - okay
Reviewer 2 Report
Comments and Suggestions for Authors
Abstract section: Modify the first lines, since sporotrichosis is not only a zoonotic infection.
I recommend modifying the keywords; staying Sporothrix
I recommend modifying the title of the manuscript since clear results on zoonosis are not shown.
Introduction section: Rearrange the ideas from lines 27-29
Fungal Culture.
I consider that the strongest part of the research is the clear identification of the pathogen. Therefore, I strongly suggest that a molecular characterization of the isolates be performed (there are some molecular markers that can help you).
Microbiological examination seems to me to be a weak point of the research. It could also be accompanied by more data from the literature showing microscopic images of both the conidia and the yeasts of the fungus.
Also, normally S. brasiliensis is a pathogen most commonly found in felines, however, you propose that it is S. schenckii. Once again, I think they should strengthen this point.
Regarding zoonosis, were the patients diagnosed? Or were they only redirected to the health center?
Fig. 5 needs more clarity, for example, it does not have metric bars, there is no comparison with the literature, and in panel c the yeasts are not observed, which needs to show a micropreparation.
Author Response
Abstract section: Modify the first lines, since sporotrichosis is not only a zoonotic infection - okay
I recommend modifying the title of the manuscript since clear results on zoonosis are not shown - One of the reviewers suggested the same thing. I think you're right and I'll modify accordingly;
I recommend modifying the keywords; staying Sporothrix - okay
Introduction section: Rearrange the ideas from lines 27-29 - okay
It was my intention to perform molecular techniques; however, our research was not funded, so unfortunately, we couldn't afford it.
I have added a couple of paragraphs strengthening the point of the microscopic image since we do not have molecular results. Moreover, I changed to Sporothrix spp since we can't be sure without the PCR.
Regarding the zoonosis, one of the patients was diagnosed, but I did not have access to their lab exams (I'm not sure if they exist at all). I directed the other ones to a health center but couldn't follow up on them. Do you think I should remove this data, anyway?
My colleague took those pictures. I'm afraid I won't have different pictures to replace them. What can I do to make the pictures clearer? Or should I remove them? Moreover, we don't have câmeras with metric bars in my university. Is there any software I can use?
Reviewer 3 Report
Comments and Suggestions for Authors
Dear authors
I think this is a really interesting article that points to the out the importance of considering that in addition to the hyperendemic areas of sporotrichosis, there are already outbreaks caused by S. brasiliensis in northeastern Brazil.
I found only small errors:
1. Line 78. Fine is repeated
2. Line 86. You make the identification of S. schenckii sensu lato by morphology
3. Line 111. Do you mean three years or three months?
4. Table 2: Correct No in raw corresponding to patient M7
5. Figure 4: Image D is not good enough to see what you mention in line 155-156
6. Line 200. Correct similar (it says simar)
7. Line 206. Correct sporothricosis and write it without capital letter
Author Response
I found only small errors:
- Line 78. Fine is repeated - ok
- Line 86. You make the identification of S. schenckii sensu lato by morphology - That's right. I have rephrased it.
- Line 111. Do you mean three years or three months? - Three months. Thank you for pointing that out
- Table 2: Correct No in raw corresponding to patient M7 - okay
- Figure 4: Image D is not good enough to see what you mention in line 155-156 - I'm afraid that is the best image I have. I added some circles so the hyphae will be easily identifiable
- Line 200. Correct similar (it says simar) -okay
- Line 206. Correct sporothricosis and write it without capital letter - okay
Thank you for your kind review
Reviewer 4 Report
Comments and Suggestions for Authors
This work provides an interesting and significant public health report on an outbreak of feline sporotrichosis in an area of the state of Alagoas, Brazil, where this disease has previously been uncommon. This report is of considerable interest, especially considering that sporotrichosis is becoming a public health problem not only for wild and domestic animals but also for humans and caregivers. The association with vulnerability and inadequate reporting systems may cause this zoonosis to be more neglected and, therefore, more difficult to control. Hence, the publication of data such as this article is valuable for informing stakeholders and providing information on current trends, particularly given that the spread of pathogenic and emerging infectious diseases in animals often precedes their expansion in humans.
The methodological approach of this report is sound, with comprehensive epidemiological data provided on the feline subjects and well-presented graphical content to complement the report. However, there are some flaws that I would like to address.
The most significant flaw in this article is the claim of documented “zoonotic transmission” of the outbreak, while only true cases were described in the animals. No information is given about the lesions of the owners, their clinical and microbiological diagnoses, or their clinical outcomes. While it is stated that “satellite lesions” appeared in three owners bitten or scratched by the animals, which can be assumed to be of the same infectious origin, the absolute claim that this is zoonotic transmission cannot be made. I suggest the authors refer to “potential or possible zoonotic transmission.” For example, in the title: “Outbreak of feline sporotrichosis with possible zoonotic transmission in the Seventh Health District of Maceió-AL,” and subsequently in the article introduction, results, or discussion. Alternatively, they could delve further into the subsequent diagnosis and provide information on the clinical history and specific data on microbiological isolations in the human patients. This is the main major flaw for me. It does not affect the validity of the animal reports but is importantly misleading and dampens my enthusiasm when reading the abstract and title of the article.
Additionally, the identification of the fungus is stated to be performed microscopically by morphological keys, identified as S. schenckii. However, I believe more information could be provided. The genus Sporothrix comprises several species, with the medically important species primarily belonging to the Sporothrix schenckii complex. S. schenckii sensu stricto and S. brasiliensis are possibly the most important, and I believe their differentiation is of clinical and epidemiological importance, especially considering that S. brasiliensis is increasingly identified in Brazil and other parts of South America. Although not directly shown, feline and zoonotic outbreaks have been hypothesized to be related to gradual temperature increases that have enabled its adaptation to invasive yeast growth.
In this regard, I suggest clarifying the nomenclature in the article (“Sporothrix sp.”, “Sporothrix”, “Sporothrix schenckii” are stated differently throughout the manuscript). It would be interesting to specify if species identification has been performed or if molecular techniques such as PCR have been used, due to the importance of distinguishing between S. schenckii sensu stricto and S. brasiliensis. If not, I suggest maintaining consistent nomenclature, such as “Sporothrix spp.” or “S. schenckii group species.”
Furthermore, providing some information on the hot topic of climate change impacts on this dissemination and virulence could be of interest. See: Etchecopaz A, Toscanini MA, Gisbert A, Mas J, Scarpa M, Iovannitti CA, Bendezú K, Nusblat AD, Iachini R, Cuestas ML. Sporothrix Brasiliensis: A Review of an Emerging South American Fungal Pathogen, Its Related Disease, Presentation and Spread in Argentina. J Fungi (Basel). 2021 Feb 26;7(3):170. doi: 10.3390/jof7030170. PMID: 33652625; PMCID: PMC7996880.
There are several minor style mistakes:
- Unify "Seventh Health District" in capital letters or not.
- Correct referencing style as per journal guidelines. Sometimes references are cited in brackets, sometimes in superscript.
- Some double spaces in the text between sentences, e.g., lines 45 and 166. Lack of spaces in other places: line 197.
- Numbers are inconsistently stated in letters or numerals. Please unify, e.g., lines 108-110.
- Some errors to correct: e.g., “being bitter” line 184, “similar results” line 200, “Sporotrichosis” line 206.
Comments on the Quality of English LanguageThe quality of the English language in this manuscript is okay but needs improvement. There are several grammatical errors, inconsistent use of terminology, and stylistic issues that hinder clarity. For instance, terms like "Seventh Health District" should be consistently capitalized, and referencing styles need to follow the journal's guidelines. Additionally, the manuscript contains typographical errors, such as double spaces between sentences and inconsistent formatting of numbers. Addressing these issues will enhance the readability and professionalism of the report.
Author Response
Regarding the zoonotic transmission, I deeply respect your expertise and agree with your points. I found it intriguing to include this data; as you rightly pointed out, the spread of pathogenic fungi in animals often precedes their expansion in humans.
However, I don't have any more information on these human patients. Two of them told me about the diagnosis but did not share the laboratory tests (I wonder if they exist). The other one told me about being bitten by the cat, and I sent her to a referral hospital, but I was not able to follow up on the case.
What do you think should be my approach? Should I remove this data altogether? Should I only mention that owners were bitten and scratched and then talk about the zoonotic potential in the discussion?
I also agree with changing the title of the article accordingly.
I"ll change the nomenclature to Sporothrix sp. It was my intention to perform molecular tehcniques, but our research was not fundend, so we weren't able to do it.
I have questions about how to state numbers since English is not my first language.
Numbers 1-9 - I have written out the words.
Numbers 10 and above - I have used numerals.
Measures I have used numerals
Is that correct?
I also have adressed the minor mistakes you noticed it. Let me know if anything went unnoticed.
Round 2
Reviewer 2 Report
Comments and Suggestions for Authors
There are some typographical errors regarding the name of the microorganisms.
You need to put the metric bar on the micrographs.
Restructure the conclusion.
Author Response
There are some typographical errors regarding the name of the microorganisms - okay
You need to put the metric bar on the micrographs - okay
Restructure the conclusion - okay
